# Overweight, obesity, and thinness among a nationally representative sample of Norwegian adolescents and changes from childhood: Associations with sex, region, and population density

**Bente Øvrebø**[1,2,3]*, **Ingunn H. Bergh**[2], **Tonje H. Stea**[4], **Elling Bere**[1,2,3], **Pål Surén**[5], **Per M. Magnus**[6], **Petur B. Juliusson**[7], **Andrew K. Wills**[8]

**1** Department of Sport Science and Physical Education, University of Agder, Kristiansand, Norway, **2** Department of Health and Inequalities, Norwegian Institute of Public Health, Oslo, Norway, **3** Centre for Evaluation of Public Health Measures, Norwegian Institute of Public Health, Oslo, Norway, **4** Department of Health and Nursing Sciences, University of Agder, Kristiansand, Norway, **5** Department of Child Health and Development, Norwegian Institute of Public Health, Oslo, Norway, **6** Center for Fertility and Health, Norwegian Institute of Public Health, Oslo, Norway, **7** Department of Health Registry Research and Development, Norwegian Institute of Public Health, Bergen, Norway, **8** University of Bristol, Bristol, United Kingdom

* bente.ovrebo@uia.no

## Abstract

### Objective

To estimate the prevalence of overweight, obesity, and thinness among Norwegian 13-year-olds and the changes from childhood (age 8 years) to adolescence (age 13 years); and to explore associations with sex, region, and population density from childhood to adolescence.

### Design

We used longitudinal, anthropometric data collected by school health nurses conducted in Norway. Weight status was classified according to the International Obesity Task Force cut-offs for overweight, obesity, and thinness, and according to mean body mass index ($kg/m^2$).

### Participants

The Norwegian Youth Growth Study, consisting of a nationally representative sample of Norwegian 13-year-olds (n = 1852; 50.7% girls), which is a part of The Norwegian Growth Cohort.

### Results

Among 13-year-old Norwegians, the prevalence of overweight (including obesity), obesity, and thinness was 15.8%, 2.5%, and 7.3%, respectively. There was little evidence that these had changed from 8 to 13 years. From 8 to 13 years, the odds of obesity was highest in the

Regulation as it contains personal data which is potentially identifying participant data. The Norwegian Regional Committees for Medical and Health Research Ethics, department South-East, has approved this study. The Personal Data Regulations and Health Research Act (§ 7) in Norway and general data protection regulation (GDPR) in the EU restrict sharing of these data. The Norwegian Institute of Public Health administer the data used in this study. External researchers can apply for access to indirectly identifiable data based on appropriate legal bases for processing the data in accordance with GDPR Article 6(1) and 9(2). To apply for access to the data for non-Norwegians: https://www.fhi.no/en/more/access-to-data/applying-for-access-to-data/. For more information about access to the data used in this study or questions regarding data requests: https://www.fhi.no/div/helseundersokelser/vekstkohorten/tilgang-til-data-fra-vekstkohorten/#soek-om-datatilgang (website in Norwegian) or contact vekstkohorten@fhi.no.

**Funding:** This work was supported by the Norwegian Research Council (grant number 260408/H10). The Norwegian Research Council had no role in the design, analysis or writing of this article.

**Competing interests:** The authors have declared that no competing interests exist.

Northern region of Norway compared to the South-East (odds ratio (OR): 3.78 (95% confidence interval (CI): 1.13, 12.65; p = 0.036) and in rural areas (OR: 4.76 (95% CI: 1.52, 14.90; p = 0.027). Over the same age period, girls had a trend towards a higher odds of thinness compared to boys (OR: 1.65 (95% CI: 0.98, 2.78; p = 0.057).

## Conclusions

In Norway, the prevalence of overweight, obesity, and thinness among 13-year-olds seem to be established by age 8 years. The prevalence of obesity was higher in the North and in rural areas. The results indicate the continued need for early prevention and treatment, and targeted interventions to certain areas.

## Introduction

The global trends of rising body mass index (BMI) among children and adolescents has leveled off in many high-income countries including Norway, however, it remains high [1]. Overweight and obesity during childhood represent a severe public health problem that negatively impacts psychosocial and cardiovascular health [2], and has a strong link to adult obesity with increased morbidity and mortality [2,3]. Understanding the population prevalence of overweight and obesity in childhood and adolescence is therefore important.

To date there are no nationwide population-based and distribution wide estimates on weight status among Norwegian adolescents between the age of 9 to 15 years. One national cross-sectional survey from 2015 reported that 15% of 8-year-old children were overweight or obese (ov/ob), and that the secular trend had plateaued in the period between 2008 to 2015 [4], but with geographical differences showing higher prevalence of ov/ob in the Northern region of Norway [5], and in rural areas compared to urban areas [6,7]. Recent regional findings from Mid-Norway report an increase in obesity among adolescents aged 13–19 for both girls and boys, in the period from 2006–2008 to 2017–2019 [8], but we have no information about changes from childhood to young adolescence.

Thinness is associated with different adverse health consequences, such as nutritional deficiencies, impaired growth and development among children, menstrual irregularity in girls/women, decreased cognitive and work capacity, and increased risk of infections [9,10]. Few studies report thinness among Norwegian adolescents, and existing evidence from European countries show variations in prevalence and trends [11]. Development of body dissatisfaction among children [12] and adolescents [13], and the increases in anxiety disorders among adolescent girls [14], with the further potential negative consequences of thinness [9,10], emphasize the importance of reporting the prevalence of thinness.

Updated information across all weight categories is essential to develop, implement, and evaluate policies and preventive measures. There is a need for up-to-date information on the prevalence of overweight, obesity, and thinness among young Norwegian adolescents. Knowledge is also lacking about the population changes in BMI and weight status from childhood into adolescence, for the same subjects. In addition, it is important to describe the occurrence of obesity and thinness according to place of living and whether such patterns are changing as children move into adolescence.

The Norwegian Youth Growth Study (NYGS) is a national cross-sectional study of 13-year-olds (8th graders), recruited in 2017 with measurements of height and weight at the age of 13 years, for whom there is also available data from the same participants at the age of 8 years.

Our aims were to (1) estimate the prevalence of overweight, obesity, and thinness, and the mean BMI at 13 years; (2) assess whether this has changed from childhood (age 8 years) to adolescence (age 13 years); and (3) describe associations with sex, region, and population density from childhood to adolescence (for example, are there patterns by region? are they widening from childhood to adolescence?).

## Methods

To estimate the prevalence of overweight, obesity, and thinness, we used longitudinal height and weight data from the NYGS consisting of a national cross-sectional sample of 13-year-olds born in 2004 and recruited from school 8th graders in 2017. The NYGS is a part of The Norwegian Growth Cohort. We also used height and weight measurements from these participants recorded by the School Health Service as a part of the routine measurements in the 3rd grade (age 8 years), offered to all Norwegian children. The NYGS was conducted by the Norwegian Institute of Public Health in collaboration with the School Health Service.

### Study design

The objective of the NYGS was to obtain nationally representative information on height and weight among school children. The NYGS used a two-stage sampling design, stratified by region (North; Mid; West; and South-East). In the first stage, ten out of 19 counties were sampled. These were selected to ensure geographical coverage of Norway over the four strata. Sampling was designed to be self-weighted at the regional and county level. Regions with smaller population sizes were oversampled to enhance precision. In the second stage, schools were randomly sampled with probabilities proportional to school size. Both private and public schools were sampled as the distinction between schools in Norway is modest.

Additionally, only schools with ten or more 8th graders were included, and only one 8th grade class was selected to participate. Further, the maximum number of participants was also limited to 30 per school due to limitations in resources. Sampling weights were calculated that additionally account for the sampling at the school level.

### Recruitment and participants

A total of 159 secondary schools (with 8th to 10th grade) were invited. The managing nurse of the School Health Service responsible for one or more of the 159 schools was contacted and asked if the Service would be willing to participate in the study, and to measure and report 8th graders' height and weight. Four schools refused participation. These were replaced by four schools from a substitute list, three of which accepted. During recruitment, a further eight schools withdrew, leaving 150 participating schools. Valid consent was received from the parents of 1907 8th graders.

### Anthropometry

The school health nurses measured height and weight of the 8th graders in the fall of 2017, one nurse at each school. The nurse asked the adolescent to remove loose objects from pockets/hair, wear light indoor clothing and no shoes, and measured weight using a digital weight scale and height with a fixed height measuring device, according to national guidelines [15]. Weight was recorded to the nearest 0.1 kg and height to 0.1 cm. Additionally, the school health nurses also reported 3rd grade height and weight measurements (when participants were approximately 8 years old—available for 1478 of the 1907 8th graders) from the adolescent's health card. The previous measurement recorded by the School Health Service was also conducted in

accordance with the same national guidelines. All measurements were entered by nurses onto an electronic questionnaire and cleaned using a longitudinal algorithm that checked for logical errors and internally inconsistent values (see details in S1 Text) [16].

## Variables

The main outcome variables were BMI z-scores, and age- and sex-specific cut-off in BMI values for overweight, obesity, and thinness. The International Obesity Task Force (IOTF) BMI cut-offs and IOTF LMS parameters were used to categorize BMI and calculate BMI z-score, respectively [17]. In the current paper, we use the terminology severe obesity instead of morbid obesity. We used inclusive IOTF categories unless otherwise stated so overweight includes obesity and severe obesity (referred to as ov/ob in the text), and similarly for categories of thinness. The IOTF cut-offs link BMI values at 18 years to centiles in childhood, giving: overweight: BMI $\geq$25 kg/m$^2$; obesity: BMI $\geq$30 kg/m$^2$; and severe obesity: BMI $\geq$35 kg/m$^2$. Similarly, we used the following World Health Organization (WHO) cut-offs for thinness: thinness grade 1: BMI <18.5 kg/m$^2$; thinness grade 2: BMI <17 kg/m$^2$; and thinness grade 3: BMI <16 kg/m$^2$ [18].

The school health nurse also completed an electronic school questionnaire with information about the school, the number of girls and boys in the 8$^{th}$ grade, and the number of pupils with signed consent. This background information was received from 137 of the 150 participating schools.

Norway is divided into four health regions (North; Mid; West; and South-East). We used this division to describe the outcomes by geographical region. Population density was categorized into three categories based on information from the schools the adolescents were attending, provided by Statistics Norway: urban (municipalities with a population >50000); semi-urban (municipalities with a population between 15000–50000); and rural (municipalities with a population <15000).

## Ethics

This study was conducted according to the guidelines laid down in the Declaration of Helsinki and all procedures involving research study participants were approved by the Norwegian Regional Committee for Medical Research Ethics (2017/431). Written informed consent was obtained from all participants and their parents, including consent for the school health nurses to collect previous routine height and weight measurements from school health records. Researchers were provided with de-identified data provided by the Division of Health Data and Digitalisation at the Norwegian Institute of Public Health, which could only be accessed through a secure platform in compliance with the Norwegian privacy regulations.

## Statistical methods

The sample was first described in terms of age, sex, and demographic characteristics. We also describe the height of children using the international WHO growth reference [19]. Plots of individual values with IOTF cut-offs and centiles overlaid were also produced to visualize the individual values.

The prevalence of IOTF overweight, obesity, severe obesity, and thinness at 13 years were estimated along with the mean BMI IOTF z-score. The sampling design was incorporated into these estimates.

To estimate the change in IOTF overweight, obesity, and thinness (grade 1 and grade 2) from childhood (8 years) to early adolescence (13 years) and associations with sex, region, and population density, multilevel logistic models with a random intercept were fitted using

maximum likelihood with an adaptive Gauss Hermite quadrature integration method and 25 integration points. Exclusive categories of grade 2 and 3 thinness were not included in the analysis of associations with sex, regions and population density due to the small case numbers. The models were parameterized to allow exposure to explain the log odds of the outcome at age 8 years and the change in log odds between 8 and 13 years, thereby assessing whether an association was present at 8 years and whether there was evidence that the association differed from 8 to 13 years. The latter was tested by including an interaction term with age. Models were also fitted without the age-interaction, and results from both sets of models are presented. Where there was little evidence of an interaction with age and any level of a factor (p>0.1), we interpret the more parsimonious model where associations are assumed to be the same at 8 and 13 years. Generally, findings were similar regardless of including the age interaction, and since only 1/21 comparisons were indicative of an age-interaction, we present the most parsimonious models in the main text and the age-interaction models in the supporting information. Similar sets of multilevel models were fitted to the continuous IOTF z-score outcome.

We were also interested in disentangling associations of region and population density with IOTF outcomes, but these are strongly correlated, and the sample size was insufficient for the categorical outcomes. However, we were able to mutually adjust for these factors using the continuous BMI IOTF z-score outcome.

Lastly, to facilitate interpretation, we also plot the predicted marginal prevalences and means by each level of each factor at each age. Stata version 16.1 software (StataCorp. 2019. Stata Statistical Software: Release 16. College Station, TX: StataCorp LLC) was used for all analyses.

## Results

### Sample description

Fig 1 describes how the analysis sample of adolescents was derived from recruitment for the prevalence and longitudinal analysis. Less than 4% were omitted due to age eligibility or missing or erroneous data (Fig 1). The prevalence estimates at age 13 years were based on 1838/1907 children while the longitudinal change models contained 1852/1907 children with at least one measure at the 8- or 13-year assessment (1467/1852 had both measures).

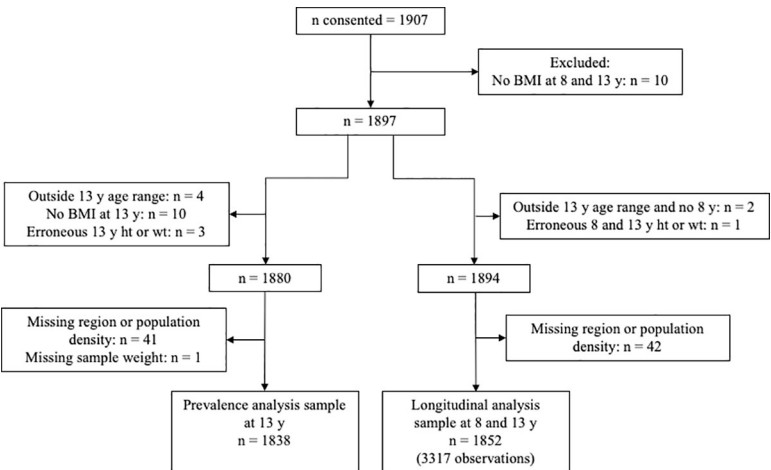

**Fig 1. Description of how the analysis sample of adolescents was derived from recruitment for the prevalence and longitudinal analyses.** BMI, Body mass index; ht, height; wt, weight; y, years.

The cohort was evenly split by sex, and the median age at the 3<sup>rd</sup> and 8<sup>th</sup> grade assessment was 8.4 and 13.4 years respectively (Table 1). As expected, given the sampling design, adolescents were evenly spread across regions. Most children (60%) resided in urban dwellings, with approximately 15% living in rural areas (Table 1). Region and population density were strongly correlated; in the South-East and West of Norway, most schools were located in urban areas (95% in the South-East and 80% in the West) whereas in Mid-Norway and the North, most schools were in semi-urban and rural areas (approximately 60% in Mid-Norway and 80% in the North) (see S1 Table). Compared to the WHO growth reference, adolescents were on average 0.5 standard deviation (SD) taller, corresponding to approximately +3.7 cm and +3.5 cm in boys and girls at 13 years respectively (see S2 Table and S1 Fig).

## Prevalence of IOTF categories at 13 years

Overall, 15.8% (95% confidence interval (CI): 13.5, 18.4%) of children were categorized as ov/ob, 2.5% (95% CI: 1.7, 3.5%) as obese and 0.3% (95% CI: 0.1, 1.0%) as severely obese; 7.3% (95% CI: 6.0, 8.8%) had a BMI below the grade 1 thinness threshold; 1.2% (95% CI: 0.7, 2.0) had grade 2 thinness and 0.1% (95% CI: 0.02, 0.9%) had grade 3 thinness (Table 2). The mean BMI IOTF z-score was 0.32 (95% CI: 0.27, 0.36). The prevalences and mean BMI by sex, region, and population density are also shown in Table 2, but associations are assessed formally in the multi-level analyses (described below). Nonetheless, at age 13 years mean BMI, overweight, and obesity was highest in the North and in rural areas, and thinness was more prevalent among girls.

For completeness, the prevalence by exclusive IOTF categories and plots of the individual values with the IOTF cut-offs and centiles overlaid are shown in supporting information (S3 Table and S2 Fig, respectively).

## Change in overweight, obesity, and thinness, and mean BMI from 8 to 13 years

Although the odds of being overweight (odds ratio (OR): 1.2; 95% CI: 0.93, 1.67), obese (OR: 1.6; 95% CI: 0.89, 3.03), thin grade 1 (OR: 1.3; 95% CI: 0.90, 1.85), thin grade 2 (OR: 1.2; 95%

**Table 1. Description of the Norwegian Youth Growth Study sample (max N = 1852)[*].**

| | | | Percentile or percent | |
| --- | --- | --- | --- | --- |
| | N | Median or n | 25<sup>th</sup> | 75<sup>th</sup> |
| Age at grade 8[†] (years) | 1839 | 13.4 | 13.1 | 13.6 |
| Age at grade 3[‡] (years) | 1478 | 8.4 | 8.0 | 8.7 |
| Female | 1852 | 938 | 50.7 | |
| Region | | | | |
| South-East | 1852 | 495 | 26.7 | |
| West | | 481 | 26.0 | |
| Mid | | 441 | 23.8 | |
| North | | 435 | 23.5 | |
| Population density | | | | |
| Urban | 1840 | 1111 | 60.0 | |
| Semi-urban | | 456 | 24.6 | |
| Rural | | 285 | 15.4 | |

N, max number that the information was derived from as included in the longitudinal analysis.

[*]Data shown in n (%) unless otherwise stated.

[†]Age at recruitment (8<sup>th</sup> grade).

[‡]Age in 3<sup>rd</sup> grade health card assessment.

**Table 2. Prevalence (%, 95% CI)\* of IOTF† overweight, obesity, and thinness among Norwegian 13-year-olds and mean IOTF z-score‡ (n = 1838).**

| | Overweight | | Obesity | | Severe obesity | | Thinness grade 1 | | Thinness grade 2 | | Thinness grade 3 | | IOTF z-score | |
|---|---|---|---|---|---|---|---|---|---|---|---|---|---|---|
| | BMI ≥25 | | BMI ≥30 | | BMI ≥35 | | BMI <18.5 | | BMI <17 | | BMI <16 | | | |
| | % | 95% CI | % | 95% CI | % | 95% CI | % | 95% CI | % | 95% CI | % | 95% CI | Mean | 95% CI |
| Overall | 15.8 | 13.5,18.4 | 2.5 | 1.7, 3.5 | 0.3 | 0.1, 1.0 | 7.3 | 6.0, 8.8 | 1.2 | 0.7, 2.0 | 0.1 | 0.02, 0.9 | 0.32 | 0.27, 0.36 |
| Sex | | | | | | | | | | | | | | |
| Boys | 15.5 | 12.5, 19.0 | 3.0 | 1.9, 4.8 | 0.3 | 0.04, 1.7 | 5.9 | 4.1, 8.2 | 1.2 | 0.6, 2.5 | 0.3 | 0.04, 1.8 | 0.35 | 0.27, 0.44 |
| Girls | 16.1 | 12.8, 19.9 | 1.9 | 1.1, 3.4 | 0.3 | 0.04, 1.9 | 8.7 | 6.6, 11.4 | 1.2 | 0.5, 2.7 | 0.0 | NA | 0.29 | 0.20, 0.38 |
| Region | | | | | | | | | | | | | | |
| South-East | 15.6 | 12.0, 20.1 | 2.2 | 1.3, 3.8 | 0.4 | 0.1, 1.8 | 7.1 | 5.2, 9.7 | 1.4 | 0.7, 2.9 | 0.2 | 0.02, 2.1 | 0.34 | 0.25, 0.42 |
| West | 14.1 | 11.3, 17.3 | 1.9 | 0.8, 4.5 | 0.0 | NA | 7.1 | 5.2, 9.7 | 0.9 | 0.3, 2.1 | 0.0 | NA | 0.29 | 0.21, 0.37 |
| Mid | 17.4 | 12.8, 23.3 | 3.0 | 1.6, 5.6 | 0.0 | NA | 7.8 | 5.1, 12.0 | 1.2 | 0.5, 2.6 | 0.2 | 0.03, 1.8 | 0.29 | 0.20, 0.38 |
| North | 18.6 | 15.2, 22.6 | 4.6 | 3.0, 7.2 | 0.5 | 0.1, 1.8 | 7.6 | 5.6, 10.4 | 0.7 | 0.2, 2.0 | 0.0 | NA | 0.35 | 0.26, 0.45 |
| Population density | | | | | | | | | | | | | | |
| Urban | 15.2 | 12.5, 18.4 | 2.1 | 1.3, 3.4 | 0.4 | 0.1, 1.3 | 7.6 | 6.0, 9.5 | 1.3 | 0.7, 2.4 | 0.1 | 0.01, 1.5 | 0.30 | 0.22, 0.38 |
| Semi-urban | 16.6 | 11.6, 23.3 | 2.9 | 1.5, 5.6 | 0.0 | NA | 6.4 | 4.1, 10.0 | 0.9 | 0.4, 2.2 | 0.0 | NA | 0.39 | 0.26, 0.52 |
| Rural | 19.0 | 14.6, 24.4 | 5.1 | 3.4, 7.4 | 0.0 | NA | 6.1 | 4.4, 8.5 | 0.8 | 0.2, 3.3 | 0.3 | 0.03, 3.1 | 0.38 | 0.28, 0.49 |

BMI: Body mass index, kg/m$^2$; IOTF, the International Obesity Task Force; NA, Not applicable due to no observations.

\*Estimates are weighted by the sampling design.

†Categories are inclusive; overweight includes obesity and severe obesity, and similarly for categories of thinness.

‡IOTF z-score: Age- and sex-specific standardized z-score calculated from the IOTF LMS parameters.

CI: 0.48, 3.2) were all higher at age 13 years compared to 8 years (adjusted for sex), there was no strong statistical evidence for these differences and the differences were relatively small on the absolute scale of change in prevalence (see S3 Fig). Similarly, the mean IOTF z-score from 8 to 13 years was also flat (mean difference (age 13 years– 8 years): +0.02 z-score; 95% CI: -0.02, 0.05).

## Associations with sex, region and population density from 8 to 13 years

Only the relationship between region and IOTF grade 1 thinness showed any evidence of having a different association between age 8 and 13 years (age interaction)–described below. For all other associations the OR was broadly similar at age 8 and 13 years.

There was no strong evidence for a difference in overweight or obesity between boys and girls (Table 3, Fig 2), but girls had a slightly higher odds of grade 1 thinness from 8 to 13 years of age. Boys also had a slightly higher mean IOTF BMI z-score at age 8 years, but by age 13 there was little evidence of a difference between the sexes (Table 4, Fig 3).

At age 8 and 13 years, the odds of obesity were highest in the North (OR: 3.78) and lowest in the West of Norway (OR: 0.62), however, there was no strong evidence for a difference in overweight (Table 3) or mean BMI between regions (Table 4, Fig 3). Thinness showed different associations with region at age 8 and 13 years; at 8 years the North and South-East had the lowest and highest prevalence, but by 13 years there was no difference in thinness between regions (see S4 Table and S4 Fig).

At age 8 and 13 years, children from rural areas had, on average, higher odds of being obese (OR: 4.76) and a higher mean BMI z-score (+0.12 z-score) compared to children from urban areas (see Tables 3 and 4 and S5 Fig). There was little evidence for a difference in thinness between children attending schools located in areas of different population density at either age 8 or 13 years.

**Table 3. Associations\* of sex, region, and population density with IOTF[†] overweight, obesity, and thinness from 8 to 13 years (n = 1852, 3317 observations).**

| | | Odds ratio | 95% CI | P value |
|---|---|---|---|---|
| **Overweight, BMI ≥25** | | | | |
| Sex | | | | |
| | Boys | Reference | | |
| | Girls | 1.23 | 0.74, 2.04 | 0.43 |
| Region | | | | |
| | South-East | Reference | | |
| | West | 0.67 | 0.33, 1.37 | 0.29 |
| | Mid | 1.16 | 0.57, 2.34 | |
| | North | 1.34 | 0.66, 2.72 | |
| Population density | | | | |
| | Urban | Reference | | |
| | Semi-urban | 1.46 | 0.80, 2.69 | 0.14 |
| | Rural | 1.92 | 0.95, 3.90 | |
| Obesity, BMI ≥30 | | | | |
| Sex | | | | |
| | Boys | Reference | | |
| | Girls | 0.68 | 0.28, 1.64 | 0.39 |
| Region | | | | |
| | South-East | Reference | | |
| | West | 0.62 | 0.16, 2.45 | |
| | Mid | 1.98 | 0.58, 6.83 | 0.036 |
| | North | 3.78 | 1.13, 12.65 | |
| Population density | | | | |
| | Urban | Reference | | |
| | Semi-urban | 1.71 | 0.58, 5.03 | 0.027 |
| | Rural | 4.76 | 1.52, 14.90 | |
| Thinness, BMI <18.5 | | | | |
| Sex | | | | |
| | Boys | Reference | | |
| | Girls | 1.65 | 0.98, 2.78 | 0.057 |
| Region | | | | |
| | South-East | Reference | | |
| | West | 0.85 | 0.43, 1.69 | |
| | Mid | 0.77 | 0.38, 1.56 | 0.80 |
| | North | 0.72 | 0.35, 1.50 | |
| Population density | | | | |
| | Urban | Reference | | |
| | Semi-urban | 0.64 | 0.34, 1.21 | 0.18 |
| | Rural | 0.56 | 0.26, 1.19 | |

BMI: Body mass index, kg/m$^2$; IOTF, the International Obesity Task Force.

\*Estimated from mixed effect logistic models that do not include an interaction with age and so the odds ratios are fixed to be the same at age 8 and 13 years.

[†]Categories are inclusive; Overweight includes obesity and severe obesity, and similarly for categories of thinness.

To try to disentangle the associations of region and population density since they are strongly correlated (see above), a mutually adjusted model was fitted (also adjusted for sex) with BMI z-score as the outcome. Mean BMI z-score was 0.13z higher (95% CI: 0.01, 0.26, p = 0.038) in children from rural areas compared to children from urban areas when adjusting

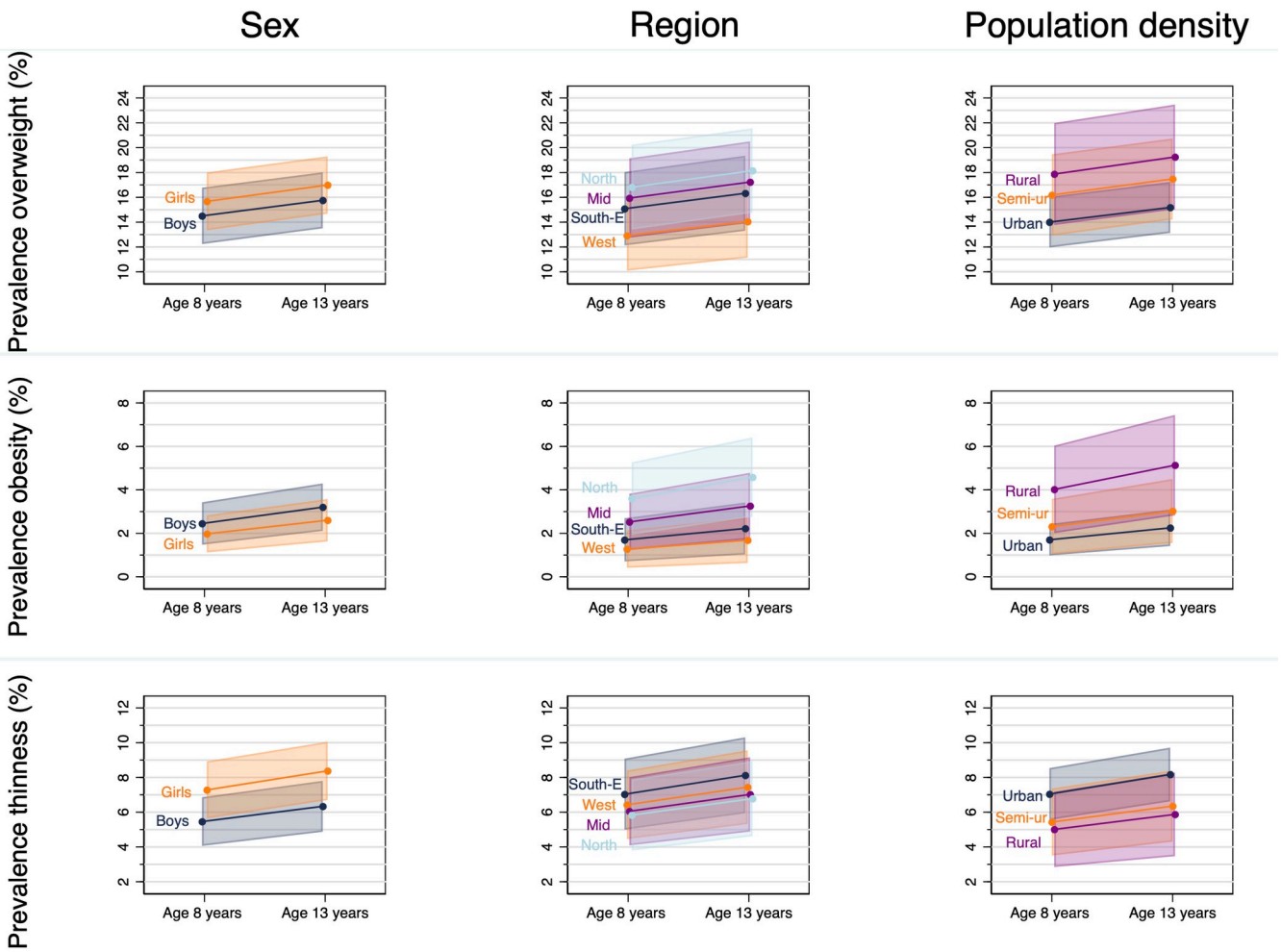

**Fig 2.** Predicted prevalence* of IOTF† overweight (top row), obesity (middle) and thinness (bottom) at 8 and 13 years, by sex (left), region (middle) and population density (right). IOTF, International Obesity Task Force. *Marginal estimates predicted as estimated from the random effect logistic regression models without an age interaction term, n = 1852, 3317 observations. The dots are the point estimates and the shaded area is the 95% confidence interval. The points have been staggered and joined between ages for clarity. †Categories are inclusive; overweight includes obesity and severe obesity, and similarly for categories of thinness.

for region, i.e. suggesting the association of population density with BMI is independent of region.

## Discussion

### Key results

In this representative nationwide study of Norwegian 13-year-olds, the prevalences of IOTF ov/ob, obesity, and thinness were 15.8%, 2.5%, and 7.3%, respectively. Further, our findings suggest that the prevalences reported at 13 years were already established in the population by age 8 years. With respect to patterns by sex, region and population density, girls had a higher odds of thinness; the odds of obesity was highest in the North and lowest in the West of Norway; and children attending schools in rural areas had the highest odds of overweight and obesity, and on average, a higher BMI. These patterns were evident at 8 years and persisted through to 13 years with no strong evidence that they were widening or shrinking.

**Table 4. Associations\* of sex, region, and population density with mean BMI IOTF z-score[†] from 8 to 13 years (n = 1852, 3317 observations).**

|  | Mean difference from 8 to 13 years | 95% CI | P value |
|---|---|---|---|
| **Sex** | | | |
| Boys | Reference | | |
| Girls | -0.01 | -0.09, 0.07 | 0.90 |
| **Region** | | | |
| South-East | Reference | | |
| West | -0.05 | -0.16, 0.06 | 0.39 |
| Mid | -0.00 | -0.12, 0.11 | 0.95 |
| North | 0.03 | -0.08, 0.15 | 0.60 |
| **Population density** | | | |
| Urban | Reference | | |
| Semi-urban | 0.06 | -0.03, 0.16 | 0.20 |
| Rural | 0.12 | 0.07, 0.24 | 0.036 |

BMI, body mass index; IOTF, the International Obesity Task Force.

\*Estimated from random effect multilevel models that do not include an interaction with age and so the estimates are fixed to be the same at age 8 and 13 years.

[†]IOTF z-score: Age- and sex-specific standardized z-score calculated from the IOTF LMS parameters.

## Interpretation

The prevalence of both overweight and obesity were higher at age 13 years compared to age 8 years but the evidence for a difference was weak. Compared to 8-year-olds measured in the Norwegian Childhood Growth Study (NCGS) in 2012 (born in the same year as our sample) and in 2015 [4], the overall ov/ob prevalence was similar among the 13-year-olds reported in the current study. This, including the previous prevalence of ov/ob among 6-11-year-old children in the Bergen Growth Study from the Western part of Norway, reporting a prevalence of 17% [20], support our findings of relatively small differences in ov/ob prevalences from childhood to adolescence.

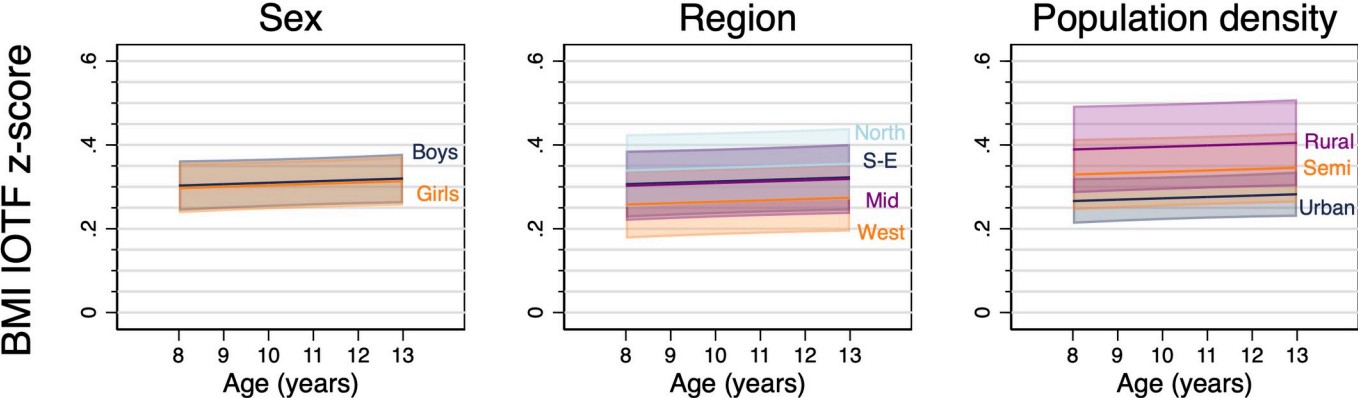

**Fig 3.** Marginal predicted\* mean BMI IOTF z-score[†] by sex (left), region (middle) and population density (right). BMI: Body mass index; IOTF, International Obesity Task Force; S-E: South-East; Semi: Semi-urban. \*Marginal estimates predicted from the random effect regression models, n = 1852, 3317 observations = 3317. The shaded area is the 95% confidence interval. [†]IOTF z-score: Age- and sex-specific standardized z-score calculated from the IOTF LMS parameters.

We were unable to detect differences in ov/ob prevalence between boys and girls, which previously have been reported among Norwegian 8-, 10-12-, and 15-year-olds, with more girls than boys having ov/ob [4,21,22]. However, other Norwegian studies have reported an opposite sex difference or no differences [4,5,20]. Sex differences could be driven by a range of various factors, which should be explored in future studies [23].

Our findings are in line with previous studies which report a lower prevalence of ov/ob in the West of Norway [20], and a higher odds of obesity in the North of Norway [5,24], and in rural areas in Norway [7] and Sweden [25]. Region and population density are strongly correlated in Norway, e.g. the Northern part of Norway contains more rural municipalities (S1 Table). Notably, definitions and characteristics of rural areas varies between countries and is here considered within a Norwegian context. We were unable to investigate associations of ov/ob with socio-economic status, however a higher prevalence of ov/ob in Northern parts of Norway and rural areas might be explained by maternal education level [6]. Yet, a previous Norwegian study reported that BMI was associated with population density independently of maternal education [7]. Furthermore, differences in prevalence by region and population density could also be explained by differences in lifestyle factors such as diet, physical activity and commuting patterns [5,26].

Compared to results from European children aged 2–13 years from 2011–2016 showing 21.3% overall ov/ob- and 5.7% obesity rates [27], our results are lower. However, there are large variations between European countries. We report only slightly higher ov/ob prevalence than the countries with the lowest ov/ob prevalence among 7-13-year-olds in 2011–2016, whereas southern European countries report a much higher prevalence of ov/ob (>30%). Prevalence of ov/ob among children aged 8–11 years from Spain are higher compared to Norwegian numbers (approximately 25% versus 16%, respectively), yet they also report a relatively steady prevalence in the period 1992 to 2017 [28]. Contrasting this, prevalence of overweight has decreased among French children (aged 4–12 years) from 2013 to 2017, while the prevalence of obesity was stable [29].

Compared to our findings, a slightly lower prevalence of thinness has been reported among 13-19-year-olds in Mid-Norway (approximately 7.3% vs. 5.5%) [8]. From 8 to 13 years, girls in the current study had a higher odds of thinness than boys. A higher prevalence of thinness among girls has previously been reported in younger children [30], however a recent report reveal a similar prevalence among 13-19-year-olds [8].

We report a prevalence of thinness of 7.3% which is markedly different from French children for whom a 25% rate of underweight among 10-12-year-olds have been reported (with an increase from 2013 to 2017) [29]. From 1992 to 2017 an increase in thinness among children in Spain has also been reported [28], yet the prevalence of thinness among 8-11-year-olds of 12% is still lower than the reports from France. In the current study, the majority of individuals categorized as thin (6.3%) fell within thinness grade 1. Cole et al. (2007) propose that a BMI of 17 (thinness grade 2 and 3) is a suitable cut-off to use as the basis for an international definition of thinness in children and adolescents [18]. Based on this, our results indicate a very low level of thinness among Norwegian 13-year-olds (with a prevalence of 1.0%). This rate (for thinness grade 2) corresponds with Sweden [31] and the Netherlands [32], and is a bit lower than prevalence from Spain (2.5%) [28].

## Implications for public health and future work

Although previous studies may have indicated a possible stabilization in secular trends of ov/ob, the prevalence of ov/ob in Norway is already high at age 8 years compared to prevalence in previous decades (1993–2000) [33], and seem to remain at this level in adolescence as shown

in the current study. Thus, policymakers should strengthen the prevention and the health service should increase early treatment of ov/ob among children and adolescents.

Our findings also suggest a need to pay attention to the development of adiposity among youth in particular parts of Norway, such as the Northern region and rural areas. This implies that targeted intervention may be needed. However, it is important to note that population density and region are strongly correlated in Norway, and while our analysis of continuous BMI suggested that population density may act independently of region, we were unable to disentangle these associations for overweight and obesity outcomes due to sample size. Future studies with much larger sample sizes and additional data on factors such as parental education level, living situation, ethnicity, and lifestyle factors, etc. are needed to understand patterns associated with weight status to define potential target populations. Furthermore, to investigate factors associated with more severe levels of thinness (grade 2 and 3), substantially larger sample sizes are needed. Additionally, studies following participants from childhood into adulthood are required to investigate the sex pattern more closely. We suggest continuing to report the prevalence of thinness to gather more knowledge on the subject, especially as there have been reported increases in Norwegian girls aged 15–17 years diagnosed with eating disorders [34].

Our work shows how BMI and overweight, obesity and thinness is non-significantly different from 8 to 13 years in the same group of children but in an aggregate population sense. Further work is needed to understand how the patterns emerge earlier in childhood, and how these indicators track in an individual sense.

## Strengths and limitations

Our results are from a nationally representative study of objectively measured height and weight. Data were checked and cleaned using a rigorous and systematic screening algorithm [16]. For analyzing age-related trends, the longitudinal design is a particular strength, and the use of multilevel models make fewer assumptions about the missing data at each age. There are, however, a few limitations. First, we needed consent from both parents, which meant we were unable to collect data from some of the 13-year-olds. However, we assessed the impact of this by comparing measurements at 8 years with a cohort from the NCGS born the same year, which did not need consent from both parents, and the prevalences were similar (results available on request). Still, we lacked information about other factors, and were unable to check if the cohorts for example were comparable regarding socio-economic status. Further, due to the design of the study, we are unable to know the total number of eligible adolescents from the schools, and therefore are unable to calculate participation rate. Secondly, although the measuring of height and weight followed national guidelines, we could not correct for clothing, which would cause a slight over-estimation of BMI. Third, when describing age-related patterns, datasets containing more than two measurement points are more ideal. Fourth, our estimates were not sufficiently precise enough to draw any conclusions with respect to whether differences across age, and also across age by region and population density are diverging, although if they are, our study suggests this is likely to be small.

## Conclusion

The prevalences of overweight, obesity, and thinness among Norwegian 13-year-olds and differences seen between regions and levels of population density seem to be largely established earlier in childhood. These findings indicate the need for both early prevention and targeted intervention, and for additional reporting of thinness in studies of BMI.

## Supporting information

**S1 Fig. Scatterplots of individual values of height in boys and girls at the 8[th] grade (age 13 years) assessment.**
(DOCX)

**S2 Fig. Scatterplots of individual values of BMI in boys and girls at the 8[th] grade (age 13 year) assessment.**
(DOCX)

**S3 Fig. Overall predicted prevalence of IOTF thinness, overweight, and obesity at 8 years and 13 years, and mean BMI IOTF z-score.**
(DOCX)

**S4 Fig. Predicted prevalence of IOTF overweight, obesity and thinness at 8 years and 13 years, by sex, region, and population density.**
(DOCX)

**S5 Fig. Marginal predicted mean BMI IOTF z-score by sex, region, and population density.**
(DOCX)

**S1 Table. Cross tabulation of region and population density, n (%).**
(DOCX)

**S2 Table. Median, 2.5[th] and 97.5[th] centiles (i.e., 95% reference range) of WHO height z-scores at 13 years (n = 1838).**
(DOCX)

**S3 Table. Prevalence (%, 95% CI) of exclusive IOTF overweight, obesity, and thinness at 13 years (n = 1838).**
(DOCX)

**S4 Table. Associations of sex, region, and population density with IOTF overweight, obesity, and thinness from 8 to 13 years (n = 1852, 3317 observations).**
(DOCX)

**S5 Table. Associations of sex, region, and population density with BMI IOTF z-score from 8 to 13 years (n = 1852, 3317 observations).**
(DOCX)

**S1 Text. Data cleaning.**
(DOCX)

## Acknowledgments

Jørgen Meisfjord contributed with expertise regarding sampling, and Ingvild Bokn was in charge of data collection. We are particularly grateful to the school health nurses for collection of data and all of the participants.

## Author Contributions

**Conceptualization:** Elling Bere, Andrew K. Wills.

**Formal analysis:** Andrew K. Wills.

**Funding acquisition:** Tonje H. Stea, Elling Bere, Pål Surén, Per M. Magnus, Andrew K. Wills.

**Methodology:** Bente Øvrebø, Ingunn H. Bergh, Petur B. Juliusson, Andrew K. Wills.

**Project administration:** Per M. Magnus.

**Resources:** Per M. Magnus.

**Software:** Andrew K. Wills.

**Visualization:** Andrew K. Wills.

**Writing – original draft:** Bente Øvrebø, Ingunn H. Bergh, Andrew K. Wills.

**Writing – review & editing:** Bente Øvrebø, Ingunn H. Bergh, Tonje H. Stea, Elling Bere, Pål Surén, Per M. Magnus, Petur B. Juliusson, Andrew K. Wills.

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
