## [Decision Letter · Decision Letter 0]

4 Jun 2021

PONE-D-21-08959

Overweight, obesity, and thinness among a nationally representative sample of Norwegian adolescents and changes from childhood: associations with sex, region, and urbanicity

PLOS ONE

Dear Dr. Øvrebø,

Thank you for submitting your manuscript to PLOS ONE. After careful consideration, we feel that it has merit but does not fully meet PLOS ONE’s publication criteria as it currently stands. Therefore, we invite you to submit a revised version of the manuscript that addresses the points raised during the review process.

We look forward to receiving your revised manuscript.

Kind regards,

Sabine Rohrmann

Academic Editor

PLOS ONE

Journal Requirements:

2. In ethics statement in the manuscript and in the online submission form, please provide additional information about the patient records/samples used in your retrospective study. Specifically, please ensure that you have discussed whether all data/samples were fully anonymized before you accessed them.

Reviewers' comments:

Reviewer's Responses to Questions

**Comments to the Author**

1. Is the manuscript technically sound, and do the data support the conclusions?

Reviewer #1: Yes

Reviewer #2: Yes

2. Has the statistical analysis been performed appropriately and rigorously? 

Reviewer #1: Yes

Reviewer #2: Yes

3. Have the authors made all data underlying the findings in their manuscript fully available?

Reviewer #1: Yes

Reviewer #2: Yes

4. Is the manuscript presented in an intelligible fashion and written in standard English?

Reviewer #1: Yes

Reviewer #2: Yes

5. Review Comments to the Author

Reviewer #1: This is a well structured and well written paper describing the results of a study using sound statistic and reporting original findings. I have some minor issues.

Methods

1. Please inform whether only public schools were included in the study or also private school. In any case it should be mentioned whether private schools are relevant in Norway.

2. It is unclear whether assessment methods of the first (older) measurement of height and weight by School Health Services are comparable to those of the NYGS. Do all participants of the NYGS have 2 measurement points?

3. Please include information about ethics approval also in the methods section (“This study was conducted according to the guidelines laid down in the Declaration of Helsinki and all procedures involving research study participants were approved by the Norwegian Regional Committee for Medical Research Ethics (2017/431). Written informed consent was obtained from all participants and their parents.”

4. Assessment methods should be described more in detail if possible: who measured (how many persons) and with which devices (scale, stadiometer)

5. The definition of urbanity / rurality is somewhat arbitrary and thus hard to compare internationally. Maybe information on population density or other internationally comparable parameters could help

Results

The used wording (“increase”, “decrease”) implies that the study found trends. However, two measurement points are not enough to describe trends especially when the assessment methods may not have been identical. Talking about “differences” would be more cautious. This issue could also be included in the “limitations” section of the discussion.

Discussion

Public Health Implication: The inclusion of additional data (living situation, educational level or migration background of the parents) via record linkage would help to better define potential target populations, e.g., see https://doi.org/10.4414/smw.2017.14501

Reviewer #2: Reviewer comments on # PONE-D-21-08959 “Overweight, obesity, and thinness among a nationally representative sample of Norwegian adolescents and changes from childhood: associations with sex, region,

and urbanicity”

The topic is self is timely relevant and interesting.

Are children in 8th grade 13 or 14 years old – please check this. In some Scandinavian schooling system preschool is considered as grade 0. To me there is a mismatch.

The study design and sampling are well described, assuring population-based sampling. However, when the participation is based on both parent consents this might challenge the population-based sampling as typically the families with higher education will participate. Please add discussion on this to the limitations.

Line 142: Additionally, the school health nurses also reported 3rd grade height and weight measurements (when participants were 7-8 years old -available for 1544 of the 1907 8th graders) recorded by the School Health Service in the adolescent’s health card. Exact age of children is needed (decimal age).

Line 157: Similarly, we used the following World Health Organization (WHO) cut-offs for thinness: Thinness grade 1: BMI <18.5 kg/m2; Thinness grade 2: BMI <17 kg/m2; and Thinness grade 3: BMI <16 kg/m2 (18). Good that the authors use thinness instead of underweight. As suggested by Cole (2008) please focus on thinness grade 2 and 3, as it is impractical and confusing to define such a high number of children and adolescents having grade 1 thinness during growth.

Results

Misleading sentence: Compared to the WHO growth reference, adolescents were on average 0.5 standard deviation (SD) taller, corresponding to approximately +3.7 cm and +3.5 cm in boys and girls at 13 years respectively (see S2 Table and S1 Fig). These are out of the scope of the paper, please omit.

Line 234: 7.3% (95%CI: 6.0, 8.8%) had a BMI below the grade 1 thinness threshold and 0.1% (95% CI: 0.02, 0.9%) had grade 3 thinness (Table 2). On purpose of not, thinness grade 2 is left out. Why? Now, thinnes is written with small capital – be consistent please.

The S3 Figure presenting prevalence of overweight, thinness and obese is misleading, as obese are in a way reported twice. BMI z-score on the other side of the panel is ok.

Line 257: Only the relationship between region and IOTF thinness grade 1 showed any evidence of having a different association between age 8 and 13 years (age interaction)…–

As mentioned earlier, I suggest to combine 2 and 3 grades and leave grade 1 out.

Discussion

The authors point, that prevalence of ov/ob remains at the same level during the follow-up – based on cross-sectional observations at two time points.

With follow-up data of same individuals, you may show how many are consistently ov/ob during the follow-up – which is more informative than this. Alternatively, how many becomes ov/ob or the opposite will support your story of stable weight status. I would direct the discussion to this direction than on plain prevalence.

Line 307..the increases were small and statistical evidence was lacking… omit the sentence. Only significant changes worth mentioning.

Line 325: We were unable to detect differences in ov/ob prevalence between boys and girls, which previously have been reported among Norwegian 8-, 10-12-, and 15-year-olds, with more girls than boys having ov/ob (4, 21, 22). Any explanation for this? We see sex-differences when using WHO reference system instead of IOTF, please discuss.

Line 327: Also, some speculation on contributing factors for higher prevalence of ov/ob in rural and North Norway. Are socioecomical backgrounds or higher education level of families different in these areas compared with urban and West-Norway? Some of these speculations are presented in line 370, reorganize the Discussion and add references, please.

Line 359: The authors wrote: “Although previous studies may have indicated a possible stabilization in secular trends of ov/ob, the prevalence of ov/ob in Norway is already high at age 8 years compared to prevalence in previous decades (1993-2000), and it remains high in adolescence as shown here (31). Thus, policymakers should strengthen the prevention and the health service should increase early treatment of ov/ob among children and adolescents.”

Consider the place of the citation.

I think the authors might draw too early conclusions on this. As the children are growing at 13 years of age, especially boys have not at age 13 undergone puberty. Growth might normalise the weight. Further evidence on stable weight status is needed.

Line 388: First, we needed consent from both parents, which meant we were unable to collect data from some of the 13-year-olds. Quite strict, in Finland consent from one parent is enough!

6. PLOS authors have the option to publish the peer review history of their article (what does this mean?). If published, this will include your full peer review and any attached files.

Reviewer #1: No

Reviewer #2: No

---

## [Author Response · Author response to Decision Letter 0]

14 Jul 2021

We have revised the manuscript with additional requirements:

Detailed ethical statement: 

“This study was conducted according to the guidelines laid down in the Declaration of Helsinki and all procedures involving research study participants were approved by the Norwegian Regional Committee for Medical Research Ethics (2017/431). Written informed consent was obtained from all participants and their parents, including consent for the school health nurses to collect previous routine height and weight measurements from school health records. Researchers were provided with de-identified data provided by the Division of Health Data and Digitalisation at the Norwegian Institute of Public Health, which could only be accessed through a secure platform in compliance with the Norwegian privacy regulations.”

We have also used PLoS Endnote Style and added details about supporting information at the end of the manuscript.

---

## [Editor Report · Decision Letter 1]

22 Jul 2021

Overweight, obesity, and thinness among a nationally representative sample of Norwegian adolescents and changes from childhood: associations with sex, region, and population density

PONE-D-21-08959R1

Dear Dr. Øvrebø,

We’re pleased to inform you that your manuscript has been judged scientifically suitable for publication and will be formally accepted for publication once it meets all outstanding technical requirements.

Kind regards,

Sabine Rohrmann

Academic Editor

PLOS ONE
---

## [Editor Report · Acceptance letter]

26 Jul 2021

PONE-D-21-08959R1 

Overweight, obesity, and thinness among a nationally representative sample of Norwegian adolescents and changes from childhood: associations with sex, region, and population density 

Dear Dr. Øvrebø:

I'm pleased to inform you that your manuscript has been deemed suitable for publication in PLOS ONE. Congratulations! Your manuscript is now with our production department. 

Kind regards, 

on behalf of

Dr. Sabine Rohrmann 

Academic Editor

PLOS ONE